# Therapeutic Potential and Nutraceutical Profiling of North Bornean Seaweeds: A Review

**DOI:** 10.3390/md20020101

**Published:** 2022-01-25

**Authors:** Muhammad Dawood Shah, Balu Alagar Venmathi Maran, Sitti Raehanah Muhamad Shaleh, Wahidatul Husna Zuldin, Charles Gnanaraj, Yoong Soon Yong

**Affiliations:** 1Borneo Marine Research Institute, Universiti Malaysia Sabah, Kota Kinabalu 88400, Sabah, Malaysia; bavmaran@ums.edu.my (B.A.V.M.); sittirae@ums.edu.my (S.R.M.S.); wahidatul@ums.edu.my (W.H.Z.); 2Faculty of Pharmacy and Health Sciences, University Kuala Lumpur Royal College of Medicine Perak, Ipoh 30450, Perak, Malaysia; charles.gnanaraj@unikl.edu.my; 3Laboratory Center, Xiamen University Malaysia, Sepang 43900, Selangor, Malaysia; yoongsoon.yong@xmu.edu.my

**Keywords:** Borneo seaweeds, marine algae, nutraceuticals, therapeutic potential, anti-inflammatory, antimicrobial, antioxidant

## Abstract

Malaysia has a long coastline surrounded by various islands, including North Borneo, that provide a suitable environment for the growth of diverse species of seaweeds. Some of the important North Bornean seaweed species are *Kappaphycus alvarezii*, *Eucheuma denticulatum*, *Halymenia durvillaei* (Rhodophyta), *Caulerpa lentillifera*, *Caulerpa racemosa* (Chlorophyta), *Dictyota dichotoma* and *Sargassum polycystum* (Ochrophyta). This review aims to highlight the therapeutic potential of North Bornean seaweeds and their nutraceutical profiling. North Bornean seaweeds have demonstrated anti-inflammatory, antioxidant, antimicrobial, anticancer, cardiovascular protective, neuroprotective, renal protective and hepatic protective potentials. The protective roles of the seaweeds might be due to the presence of a wide variety of nutraceuticals, including phthalic anhydride, 3,4-ethylenedioxythiophene, 2-pentylthiophene, furoic acid (*K. alvarezii*), eicosapentaenoic acid, palmitoleic acid, fucoxanthin, β-carotene (*E. denticulatum*), eucalyptol, oleic acid, dodecanal, pentadecane (*H. durvillaei*), canthaxanthin, oleic acid, pentadecanoic acid, eicosane (*C. lentillifera*), pseudoephedrine, palmitic acid, monocaprin (*C. racemosa*), dictyohydroperoxide, squalene, fucosterol, saringosterol (*D. dichotoma*), and lutein, neophytadiene, cholest-4-en-3-one and *cis*-vaccenic acid (*S. polycystum*). Extensive studies on the seaweed isolates are highly recommended to understand their bioactivity and mechanisms of action, while highlighting their commercialization potential.

## 1. Introduction

Marine organisms have been widely used as sources of functional bioactive compounds over the years [1]. Among marine resources, seaweeds (multicellular marine algae) are well-documented natural sources of proteins, nitrogen compounds, carbohydrates as well as lipids, vitamins, minerals, pigments and volatile compounds [1,2]. Based on chlorophyll, seaweeds are divided into three groups—green algae (Chlorophyceae), red algae (Rhodophyceae), and brown algae (Phaeophyceae) [3]—and possess a wide range of biological potentials that are beneficial against several disorders such as cytotoxic, antioxidant, anti-inflammatory, and antimitotic activities [4,5]. Due to population growth, fast industrial growth, and the public’s desire for natural products, worldwide demand for seaweed products is anticipated to grow even more in the years ahead [6].

Malaysia is part of the Coral Triangle, a geographical area in Southeast Asia and the Pacific that includes the oceans close to Indonesia, Malaysia, Philippines, Papua New Guinea, Timor-Leste and Solomon Islands. The temperatures of Malaysia’s coastal waters make it ideal for the development and growth of a wide variety of seaweed species. The North Borneo region of Malaysia is one of the main seaweeds growing areas; it has a suitable environment for cultivating a diverse variety of seaweeds and is the only region of Malaysia where seaweeds are farmed commercially [7]. In North Borneo (Sabah state), seaweeds cultivation was introduced in 1978 and the primary seaweeds cultivating sites are Semporna, Kunak, Lahad Datu and Bangi [8]. In 2008, approximately 111,298 tonnes of seaweeds were cultivated; Semporna accounted for 95%, Lahad Datu 4.4%, Bangi 0.3%, and Kunak 0.3% [7]. Some of the common species of seaweeds found in North Borneo are *Kappaphycus alvarezii* (Doty) Doty, *Eucheuma denticulatum* (Burman) Collins et Harvey, *Halymenia durvillaei* Bory de Saint-Vincent, *Caulerpa lentillifera* J. Agardh, *Caulerpa racemosa* (Forsskål) J. Agardh, *Dictyota dichotoma* (Huds.) Lamouroux, and *Sargassum polycystum* C. Agardh [9]. Brief descriptions regarding North Bornean seaweeds are listed in Table 1.

## 2. Therapeutic Potential of North Bornean Seaweeds

Seaweeds are also known as sea vegetables and have been used for the treatment of various disorders [13,14,15,16]. Anti-inflammatory, antioxidant, antimicrobial, and anticancer properties as well as cardiovascular protection, renal protection, hepatoprotection, and neuroprotection are only a few of the medicinal activities of seaweed that have been previously described [9,16,17,18,19,20,21,22]. Some of the protective actions of North Bornean seaweeds are discussed below.

### 2.1. Anti-Inflammatory Activity

Inflammation is a recognized defensive mechanism evolved in high-level organisms in response to stressors that disrupt bodily homeostasis. Microbial infections, tissue stress and some traumas are examples of hazards that cause inflammation with common symptoms of fever, redness, swelling and pain [23,24]. Overproduction of pro-inflammatory cytokines including tumor necrosis factor-alpha (TNF-α), interleukin (IL) (IL-6 and IL-1), prostaglandin E2 (PGE2), nitric oxide (NO), and enhanced production of reactive oxygen species (ROS) define the inflammatory response [25]. Increased activity of inducible nitric oxide synthase (iNOS) and cyclooxygenase-2 (COX-2) is associated with increased NO and PGE2 production [26].

*K. alvarezii* has been reported to have anti-inflammatory potential in asthma-induced rats. The extract changed circulating white blood cell levels, reduced mucin synthesis, and downregulated the expression of TNF-α, IL-4, nuclear factor kappa beta (NF-κB), epidermal growth factor receptor (EGFR) and matrix metalloproteinase (MMP-9). The consumption of seaweed may be useful for asthma patients. The extract decreases bronchiole smooth muscle thickness for airflow facilitation and decreases asthmatic inflammation, lung eosinophil infiltration, and mucin production [16].

The anti-inflammatory activity of *C. lentillifera* polysaccharides has been reported to have an inhibitory impact on lipopolysaccharide (LPS)-induced HT29 colorectal carcinoma cells, lowering the overproduction of TNF-α and IL-1β, SIgA and mucin2 (related proteins), as well as decreasing TNF-α and IL-1β expression [27]. *C. racemosa* polysaccharides have been reported to have anti-inflammatory potential and activate the hemoxigenase-1 (HO-1) pathway to sustain the production of hemoxigenase-1 enzyme, crucial for the prevention of inflammation [28]. 

In murine macrophage RAW 264.7 cells, the dichloromethane fraction of *D. dichotoma* extract at a concentration of 25 ug/mL inhibited the production of NO and PGE2, followed by a reduction in the expression of inducible nitric oxide synthase (iNOS) and COX-2 proteins, and iNOS and COX-2 mRNA in a dose-dependent pattern. COX-2 and iNOS are implicated in a variety of pathological processes, including inflammation. The solvent fractions of *D. dichotoma* extracts also decrease the mRNA expression of other cytokines including TNF-α, IL-1, and IL-6 in the murine macrophage cell line [29].

Anti-inflammatory and analgesic activity from brown algae *S. polycystum* has been reported using a mouse model with the paw edema method, where the mouse paw was inflamed and the hexane fraction of seaweed extract at a concentration of 70 mg/kg b.w. was applied. The anti-inflammatory effect was measured by the decreased percentage of edema size. The hexane fractions of *S. polycystum* extract significantly reduced the edema size compared to untreated mice [17].

### 2.2. Antioxidant Activity

Antioxidant phytochemical compounds can scavenge the reactive oxygen species (ROS) and reactive nitrogen species (RNS) in the human body, and slow down or prevent the onset of oxidative stress-related diseases including cardiovascular diseases (CVDs), neurological diseases (for example, Alzheimer’s disease (AD), Parkinson’s disease, multiple sclerosis, amyotrophic lateral sclerosis (ALS), and depression), cancer (chromosomal abnormalities, DNA damage), respiratory disease (chronic obstructive pulmonary disease (COPD) and asthma), rheumatoid arthritis, delayed sexual maturation, and kidney and liver diseases [30,31,32,33]. 

Marine seaweeds from North Borneo are a good source of antioxidants [9,11,34,35]. In a study, the antioxidant potential of marine seaweeds from North Borneo was evaluated by the Trolox equivalent antioxidant capacity (TEAC) and ferric reducing antioxidant power (FRAP) methods and total phenolic content by the Folin–Ciocalteu method expressed as phloroglucinol equivalents (PGE) [9]. The antioxidation activities of the North Bornean seaweeds showed good radical-scavenging and reducing power potential. The radical-scavenging and reducing power activities of the seaweeds have been reported in *K. alvarezii* 1.63 and 225.00 TEAC and FRAP mM.mg/dry extract and 22.50 TPC mg PGE/g dry extract, *E. denticulatum* 1.54 and 153.97 TEAC and FRAP mM.mg/dry extract and 15.82 TP mg PGE/g dry extract, *H. durvillaei* 1.67 and 182.29 TEAC and FRAP mM.mg/dry extract and 18.90 TP mg PGE/g dry extract, *C. lentillifera* 2.16 and 362.11 TEAC and FRAP mM.mg/dry extract and 42.85 TP mg PGE/g dry extract, *C. racemosa* 2.01 and 355.36 TEAC and FRAP mM.mg/dry extract and 40.36 TP mg PGE/g dry extract, *D. dichotoma* 1.66 and 268.86 TEAC and FRAP mM.mg/dry extract and 35.23 TP mg PGE/g dry extract, *S. polycystum* 1.86 and 366.69 TEAC and FRAP mM.mg/dry extract and 45.16 TP mg PGE/g dry extract compared to butylated hydroxytoluene (standard) (3.84 and 615.71 TEAC and FRAP mM.mg/dry extract) [9]. Among the above-mentioned seaweeds, *C. lentillifera* has high antioxidation activities followed by *C. racemosa*, *S. polycystum*, *H. durvillaei*, *D. dichotoma*, *K. alvarezii* and *E. denticulatum*; while in terms of TP, *S. polycystum* has indicated high values followed by *C. lentillifera*, *C. racemosa*, *D. dichotoma*, *K. alvarezii*, *H. durvillaei* and *E. denticulatum* [9].

### 2.3. Antimicrobial Activity

Pathogenic microbes including bacteria, fungi, viruses and parasites are responsible for the development of various diseases that arise in the community [36]. Seaweeds with various bioactive secondary metabolites act as antimicrobial agents [37,38].

The antibacterial potential of the aqueous extraction of *K. alvarezii* against *Staphylococcus aureus* (*S. aureus*) (Rosenbach, 1884) has been examined. Administration of the extract at a concentration of 200 mg/mL resulted in an inhibition zone of 10.03 mm [37]. The ethanol extract of *K. alvarezii* was also effective against *S. aureus*, *Staphylococcus epidermidis* (*S. epidermidis*) (Winslow & Winslow, 1908), *Pseudomonas aeruginosa* (*P. aeruginosa*) (Schroeter, 1872) and *Bacillus*
*subtilis* (*B*. *subtilis*) (Ehrenberg, 1835) bacterial strain at a concentration of 30–80% (*w*/*v*) with an inhibition zone of 11.93–14.85 mm, while ethyl acetate fractions of the ethanol extract of *K. alvarezii* at a concentration of 50% (*w*/*v*) inhibited the growth of *S. aureus*, *S. epidermidis*, *P. aeruginosa* and *B. subtilis* with inhibition zones of 7.14, 19.70, 0.73 and 18.30 mm, respectively. No antifungal activity of *K. alvarezii* extract and fractions have been published against *Candida albicans* (*C. albicans*) (C.P. Robin) Berkhout and *Aspergillus niger* (*A. niger*) (Tiegh) [38]. On the other hand, fungicidal activities of *K. alvarezii* extract have been reported against *Lagenidium spp* and *Haliphthoros* fungal strains [18]. In Malaysia and Indonesia, *Lagenidium thermophilum* (*L. thermophilum*) (K. Nakam., Miho Nakam., Hatai & Zafran) and *Haliphthoros sabahensis* (*H. sabahensis*) (Y. N. Lee, K. Hatai, O. Kurata, 2017) are pathogenic to the eggs and larval stages of *Scylla*
*serrata* (Forskål, 1775) and *Scylla*
*tranquebarica* (Fabricius, 1798) (mangrove crabs) [39,40]. The ethanol extract of *K. alvarezii* inhibited the hyphal growths of *L. thermophilum* IPMB 1401 (Y. N. Lee, K. Hatai, O. Kurata, 2016) and *H. sabahensis* IPMB 1402 [18]. Regarding antiviral activity, it has been reported that red algal lectin ECA-2 obtained from *K. alvarezii* (currently known as KAA-2 of *K. alvarezii*) exhibited strong anti-influenza activity against a wide spectrum of influenza virus strains, including the newly evolving swine-origin H1N1-2009 influenza strain. The mechanism involved the direct binding of ECA-2 to the viral envelope protein hemagglutinin (HA) and inhibited influenza virus propagation [41].

The ethanol extract of *E. denticulatum* inhibited the growth of *S. aureus* with inhibition zones of 6.0–16.3 mm. Furthermore, the minimum inhibitory concentration (MIC) and minimum bactericidal concentration (MBC) values for *E. denticulatum* extract against *S. aureus* were 10% and 15%, respectively. At 10%, minimum bacterial growth was observed, the number of bacteria greatly decreased from 3.0 × 10^7^ to 1.5 × 10^2^ CFU/plate, and turbidity levels also decreased; while at 15% of the extract, no bacterial growth was noticed [42]. No antifungal activity of *E. denticulatum* extract has been reported against *Aspergillus flavus* (*A. flavus*) (Link) [43]. Sulphated polydigalactosides (carrageenans) extracted from *E. denticulatum* have been tested for in vitro antiviral activity against human herpes virus type 1 (HHV-1). Carrageenans indicated an antiviral impact by the inhibition of virus attachment and interference in a subsequent stage of the virus replicative cycle. HHV-1 viral DNA synthesis was reduced by 3 folds in cultures treated with sulphated polydigalactosides from *E. denticulatum* (0.75 mg/mL) [44].

The extracts of *H. durvillaei* has been reported to have antimicrobial effects. The presence of antimicrobial activities of *H. durvillaei* extract against pathogenic bacteria was determined using the disc diffusion method. The solvent extract of *H. durvillaei* inhibited the growth of *P. aeruginosa* (11.89 mm), *S. aureus* (12.22 mm), and *Streptococcus pyogenes* (*S. pyogenes*) Rosenbach, 1884 (10.67 mm), respectively. However, no fungicidal activity of the solvent extract of *H. durvillaei* has been reported against *C. albicans* [45].

The chloroform extracts of *C. lentillifera* were tested against Methicillin-resistant *S. aureus* (MRSA) and neuropathogenic *Escherichia coli* K1 (*E. coli* K1). Moderate antibacterial activity of 62.17% against MRSA and poor antibacterial impact against *E. coli* K1 of 12.42% were demonstrated by *C*. *lentillifera* extract at a concentration of 250 μg/mL [35]. However, no antibacterial effect from an aqueous extract of *C.*
*lentillifera* (0–128 μg/mL) was recorded against the shrimp pathogenic bacteria *Vibrio vulnificus* (Reichelt et al., 1979) Farmer, 1980, *V. alginolyticus* (Miyamoto et al., 1961) Sakazaki, 1968, *V. parahaemolyticus* (Fujino et al., 1951) Sakazaki et al., 1963 or *V. harveyi*, (Kornicker & King, 1965), as compared with positive (enrofloxacin, 128 μg/mL) and negative controls [46]. The antiviral activity of *C. lentillifera* extract was tested against White Spot Syndrome Virus (WSSV) (family Nimaviridae, consisting of a large circular double-stranded DNA genome) [46,47]. WSSV infects shrimps and is characterized by the presence of circular white patches in the cuticle of the cephalothorax and abdominal segments, with a reddish to pinkish color. In Asia and Americas, WSSV caused mass mortalities (80–100%) of cultured shrimps [48]. The administration of *C*. *lentillifera* extract yielded very good outcomes. Shrimps injected with WSSV and *C. lentillifera* (1–10 mg/mL) preincubated solutions exhibited significantly lower mortality of 0.0–6.7%, compared with the positive control (100%) (only WSSV-injected). This inhibitory effect was further confirmed by the reduction in viral loads of WSSV, and *C. lentillifera* (1–10 mg/mL) expressed significantly lower viral loads (0.00–0.79 log_10_ copies number/µg of total DNA, respectively) than the positive control (4.39 log_10_ copies number/µg of total DNA) [46]. Fungicidal activities of *C. lentillifera* against *L. thermophilum* and *H. sabahensis* have been reported as well. An ethanol extract of *C. lentillifera* inhibited hyphal growths of *L. thermophilum* IPMB 1401, *L. thermophilum* IPMB 1601 and *H. sabahensis* IPMB 1603 [18].

A chloroform extract of *C. racemosa* demonstrated antibacterial activity against MRSA and *E. coli* K1. The extract of *C. racemosa*, at a concentration of 250 μg/mL, displayed a high antibacterial effect of 97.7% against MRSA, but a weak effect of 19.90% against *E. coli* K1. A methanol extract of *C. racemosa* (250 μg/mL) also showed antibacterial activity of 61.54% and 42.91% against MRSA and *E. coli* K1 [35], respectively. *C. racemosa* has been reported to show antifungal activity against *A. flavus*. An ethanol extract of the seaweed demonstrated the strongest inhibitory power with a 30 mm diameter inhibition zone against *A. flavus* [43]. The antiviral activity of a solvent extract of *C. racemosa* was demonstrated against the Chikungunya virus (CHIKV) [49]. The virus belongs to the alphavirus genus of the Togaviridae family, an RNA virus mostly spread by bites of *Aedes aegypti* and *Aedes albopictus* mosquitoes, which cause high fever, joint pain, back pain, vomiting, headache, kidney, liver, heart disease, etc. [50]. The antiviral potential of a solvent extract of *C. racemosa* was determined based on inhibition of the cytopathic effect caused by CHIKV on African monkey kidney epithelial (Vero) cells. Chloroform, ethyl acetate, ethanol, and methanol extracts (5 to 640 μg/mL) of *C. racemosa* showed a significant inhibition effect [49].

Methanol, dichloromethane and hexane extracts of *D. dichotoma* at a concentration of 1.5 mg/disc were investigated for in vitro antibacterial and antifungal activities. The results indicated that the methanol extract inhibited the growth of *B. subtilis* (6.5 mm) and *S. aureus* (7.5 mm). The dichloromethane extract inhibited the growth of *B. subtilis* (7.0 mm), *Enterobacter aerogenes* (*E. aerogenes*) (Hormaeche & Edwards, 1960) (6.5 mm), *E. coli* (6.5 mm), *Proteus vulgaris* (*P. vulgaris*) (11.0 mm) and *Salmonella typhimurium* (*S. typhimurium*) (Loeffler, 1892) (7.0 mm), whereas the hexane extract inhibited the growth of *B. subtilis* (9.0 mm) and *S. aureus* (7.5 mm) only [51]. The antibacterial activity of the ethanol extract of *D. dichotoma* has been shown against *Salmonella typhi* (*S. typhi*), *Klebsiella pneumoniae* (*K. pneumoniae*) (Schroeter, 1886) Trevisan, 1887 and *Shigella boydii* (*S. boydii*) (Ewing, 1949) [52]. The antifungal activity of diethyl ether, methanol, and acetone extracts of *D. dichotoma* has been reported against *Mucor* sp. and *A. flavus* [52], while no antifungal activity of the solvent extracts of *D. dichotoma* (1.5 mg/disc) has been observed against *C. albicans* [51]. The antiviral activity of various fractions of *D. dichotoma* extract has been tested against herpes simplex virus (HSV) and coxsackievirus B3 (CVB3) [53]. HSV belongs to the Herpesviridae Family, has a double-stranded DNA structure, infects humans, and causes a variety of illnesses ranging from mild mucocutaneous infections to life-threatening infections, whereas CVB3 belongs to the Picornaviridae Family, has a single-stranded RNA structure, and is responsible for a wide spectrum of human diseases, from asymptomatic to deadly infections [54,55]. The antiviral properties of the fractions were recorded in terms of virus plaque inhibition on a Vero cell monolayer. The fractions indicated moderated antiviral activities against both HSV and CVB3 viruses [53].

A solvent extract of *S. polycystum* was tested against human pathogenic bacteria. The methanol extract of the seaweed resulted in the inhibition of *P. aeruginosa* (15 mm), *K. pneumoniae* (16 mm), *E. coli* (19 mm), and *S. aureus* (20 mm) [56]. However, methanol and ethanol extracts of *S. polycystum* indicated no inhibition against *B. subtilis* or *S. enteritidis.* Similarly, no antifungal activity has been observed against *A. niger* [57].

### 2.4. Anticancer Activity

Cancer is one of the main causes of mortality in the world, and many research facilities are now focusing on the development of new anticancer medicines that could improve chemotherapy treatment and reduce mortality rates [58]. The protective role of marine products, and especially seaweeds found in North Borneo, in controlling chronic diseases such as cancer has been articulated [19,59,60,61,62].

A solvent extract of *K. alvarezii* has been reported with anti-breast and anti-colorectal cancer potential. The anticancer activities were expressed with inhibitory concentration (IC_50_) value (µg/mL). An IC_50_ value of less than 100 is considered to indicate an active compound with anticancer properties. An ethanolic extract of *K. alvarezii* exhibited anticancer activity against human breast adenocarcinoma cell line MCF-7 with an IC_50_ of 75.7 µg/mL, while ethyl acetate and hexane extracts showed anti-colorectal cancer activity against human colorectal carcinoma cell line HCT-116 with IC_50_ values of 21.4 and 43.0 µg/mL, respectively [63].

The antitumor activity of *E. denticulatum* against Ehrlich carcinoma and Meth-A fibrosarcoma has been reported. Oral administration of the extract (1600 mg/kg b.w.) for 28 days resulted in the inhibition of Ehrlich carcinoma by 25% in tumor-implanted mice. Similarly, intraperitoneal administration of *E. denticulatum* extract (50 mg/kg b.w.) for 7 days resulted in the inhibition of Meth-A fibrosarcoma by 17% [19]. The anticancer activity of crude extracts of *H. durvelaei* was investigated against four cancer cell lines (A549, HT-29, PC-3, and AGS). The results indicated that administration of *H. durvelaei* extracts reduced the growth of AGS and HT-29 cell lines by 27.17% and 1.47%, respectively [61].

Oligosaccharides (β-1,3-xylooligosaccharides) obtained from *C. lentillifera* have been reported to show antitumor properties against human breast adenocarcinoma cell line MCF-7. Exposure to *C. lentillifera* oligosaccharides inhibited the growth of MCF-7 cells in a dose-dependent manner and induced apoptosis (triggered chromatin condensation and poly ADP-ribose polymerase degradation) [59].

Polysaccharide fractions (coded as CRP) obtained from *C. racemosa* have been reported to show antitumor activity in tumor-inoculated mice (H22 tumor). The results indicated that administration of *C. racemosa* polysaccharide fractions of different doses could significantly inhibit the H22 tumor. After 14 days of transplantation, the weight of tumors in mice without polysaccharide treatment increased to 1.02 g while the tumor weight in mice exposed to the polysaccharide at a dose of 100 mg/kg b.w./day by routine oral passage decreased to 0.47 g, and the tumor inhibition rate reached 53.9% [60].

A methanol extract and fractions (petroleum ether, chloroform, ethyl acetate, n-butanol, and aqueous) of *D. dichotoma* were tested against seven different cancer cell lines including HCT-116, MCF-7, HepG2, A-549, PC-3, HeLa, and CACO in 96-well plates. The extract and fractions were applied at concentrations ranging from 0.86 to 100 μg/mL. The results demonstrated that *D. dichotoma* extract and fractions displayed significant anticancer effects against several cancer cell lines in a dose-dependent manner but were generally more selective against MCF-7 and PC-3 cell lines. The chloroform fraction was the most effective in MCF-7, PC3, and CACO cells (IC_50_ = 1.93 ± 0.25, 2.20 ± 0.18, and 2.71 ± 0.53 μg/mL, respectively) followed by the petroleum ether fraction against MCF-7 and PC-3 (IC_50_ = 4.77 ± 0.51 and 3.93 ± 0.51 μg/mL, respectively) and the ethyl acetate fraction against HepG2 and CACO (IC_50_ = 5.06 ± 0.21 and 5.06 ± 0.23 μg/mL, respectively) [62].

The anticancer activity of sulphated polysaccharides (fucoidan) from *S. polycystum* at a concentration ranging from 25–150 μg/mL has been reported against human breast adenocarcinoma cell line MCF-7 via cell viability assay. Treatment with the sulphated polysaccharides indicated the highest percentage of inhibition (90.4%) against the MCF-7 cell line at 150 μg/mL with an estimated IC_50_ of 50 μg/mL [64]. In another study, aside from MCF-7 cells, treatment with *S. polycystum* polysaccharide induced apoptosis in colorectal cancer cell lines (HCT-15 cell) [34].

### 2.5. Anti-Obesity and Cardiovascular Protection

The hypocholesterolemic effects of various marine algae and algal polysaccharides have been reported. The administration of extract significantly reduced serum total cholesterol (TC), low-density lipoprotein cholesterol (LDL-C) and triglycerides (TG) [12,20].

After 16 weeks on high-cholesterol/high-fat (HCF) diets, male Sprague-Dawley rats weighing 260–300 g had significantly higher body weight (b.w.), lipid peroxidation (malondialdehyde (MDA), end-product of lipid peroxidation), plasma low-density lipoprotein cholesterol (LDL-C), plasma total cholesterol (TC), plasma triglycerides (TG), erythrocyte glutathione peroxidase (GSH-Px), superoxide dismutase (SOD) and catalase (CAT) levels. The addition of 5% *K. alvarezii* to the HCF diet dramatically decreased body weight (29.1%), LDL-C (49.3%), plasma TC (11.4%), TG (36.1%), plasma MDA level (10.7%), GSH-Px (13.49%), SOD (9.4%) and CAT (24.48%), and significantly increased HDL-C levels (55%), compared to rats fed the HCF diet only [20]. 

*E. denticulatum* played a vital role in the reduction of fat absorption by the body via inhibition of pancreatic lipase [12]. Inhibiting the absorption of dietary fat is one of the most efficient methods to manage obesity and cardiovascular risks [65,66]. An *E. denticulatum* extract at a concentration of 3.8 mg/mL showed pancreatic lipase activity inhibition with an 83% reduction [12]. 

Treatment of HCF diet rats with *C. lentillifera* was reported to show anti-obesity and cardiovascular protection activity. Supplementation with 5% *C. lentillifera* extract for 16 weeks in HCF-diet rats reduced body weight by 39.5%, significantly increased HDL-C levels by 48.7%, reduced plasma TC by 18.4%, LDL-C by 34.6% and TG by 33.7%, and lowered plasma MDA level by 9%, GSH-Px by 31.8% and CAT by 3.14%, compared to the corresponding levels in high-cholesterol-diet rats [20]. 

The exposure of induced-hypercholesterolemia and -hypertriglyceridemia rats to *S. polycystum* significantly reduced body weight gain, plasma antioxidant enzymes and plasma lipid peroxidation to levels closer to those of healthy rats. Supplementation with 5% *S. polycystum* in rats fed a high-fat diet reduced body weight by 42.6%, significantly increased HDL-C levels by 16.2%, reduced plasma TC by 11.4%, LDL-C by 22% and TG by 7.69%, and decreased the plasma MDA level by 6.8%, GSH-Px by 43.4% and CAT by 15.7%, as compared to the corresponding levels in hypercholesterolemia and hypertriglyceridemia rats [20].

### 2.6. Hepatoprotection

Millions of people die each year as a result of hepatic diseases across the world [67,68]. The spread of hepatic disorders is aided by alcohol intake, obesity, nonalcoholic fatty liver disease, viral infection and medications [69,70,71,72,73]. Oxidative stress is one of the mechanisms underlying hepatotoxicity, which occurs when there is an imbalance between the production of reactive oxygen species (ROS) and the antioxidants’ ability to scavenge them in the liver [74]. Overproduction of ROS results in the elevation of serum hepatic marker enzymes including serum glutamate pyruvate transaminase (SGPT) or alanine aminotransferase (ALT), serum glutamic-oxaloacetic transaminase (SGOT) or aspartate transaminase (AST), and alkaline phosphatase (ALP), an indication of liver damage. It also induces lipid peroxidation and alters levels of antioxidant enzymes including glutathione peroxidases (GPs), catalase (CAT), superoxide dismutase (SOD), glutathione S-transferase (GST), quinone reductase (QR), etc. [75,76,77].

The hepatoprotective activity of *K. alvarezii* ethanolic extract administered for 25 days against lead acetate-induced hepatic injury in mice has been investigated. The extract at a concentration of 800 mg/kg b.w. reduced AST, ALT and ALP levels by 15.79%, 18.52% and 16.11%, respectively, compared to lead acetate-treated mice (20 mg/kg b.w. orally once a day for 21 days). Mice administered with ethanol extract of *K. alvarezii* (800 mg/kg b.w.) also demonstrated a significant (*p* < 0.05) elevation in SOD and GPx levels by 45.94% and 18.78%, respectively, and a significant (*p* < 0.05) reduction in MDA level by 22.83%, compared with lead acetate-treated mice. Histological observations of mouse hepatic tissues treated with *K. alvarezii* ethanolic extract indicated improved hepatic cell structure, blood congestion, and fatty degeneration compared to lead acetate-treated mice [76].

A methanol extract of *C. lentillifera* demonstrated hepatoprotection against acetaminophen (n-acetyl-p-aminophenol; APAP) induced hepatic damage in juvenile zebrafish (aged 1–3 months). The administration of APAP to the control group at a concentration of 10 μM caused fish mortality; while the introduction of the methanol extract of *C. lentillifera* at concentrations of 10, 20 and 30 μg/l to tanks holding 10 μM APAP-treated groups reduced fish mortality. Histological observation by hematoxylin and eosin staining of zebrafish hepatic tissues exposed to 10 μM APAP and concurrently administered *C. lentillifera* extract indicated a reduction in hepatic necrosis, hepatocyte swelling, hepatocyte vacuolization and leukocyte infiltration in a dose-dependent manner, as compared to the control group treated solely with 10 μM APAP [78].

An aqueous extract of *C. racemosa* at a concentration of 200 mg/kg b.w. was administered for 30 days on a daily basis to 40% carbon tetrachloride (CCl_4_) induced hepatic fibrosis rats (for 30 days, rats were given 2 mL/kg b.w. of CCl_4_ intraperitoneally on alternate days). Intoxicated rats treated with water extracts of *C. racemosa* showed significant (*p* < 0.05) decreases in their high levels of AST (46.7%), ALT (82.2%), ALP (41.3%), LDH (25.8%) and total bilirubin (69.6%), as compared to CCl_4_ treated control rats [79]. 

The protective effect of a solvent extract of *S. polycystum* was examined in acetaminophen (single dose administered intraperitoneally, 800 mg/kg b.w.) induced hepatic oxidative injured rats. The oral administration of *S. polycystum* extract in intoxicated rats at a concentration of 200 mg/kg b.w./day for 15 days reduced elevated levels of ALT (27.64%), AST (56.43%), LDH (43.38%), ALP (72.53%) and MDA (31.50%), compared to the levels in an APAP-administered control group [22].

### 2.7. Neuroprotection

Neuroprotection is a strategy for halting the progression of neurodegeneration [80]. Neurodegeneration is a multifaceted, complicated process that results in the death and loss of neuronal structures in the nervous system. Oxidative stress, calcium dysregulation, axonal transport deficiencies, mitochondrial dysfunction, inappropriate neuron-glial interactions, DNA damage and neuroinflammation are all underlying processes in neurodegeneration [80]. Seaweeds from North Borneo are rich in sulphated and non-sulphated polysaccharides and have been reported to have neuroprotection potential [21,81,82].

It was reported that an extract of *K. alvarezii* might be beneficial as a food supplement or medication for those who are prone to neurological disorders. In primary cultures of hippocampal neurons, the effects of *K. alvarezii* extracts on the development and complexity of neuronal cytoarchitecture were reported. A solvent extract of *K. alvarezii* with an optimal concentration of 1 μg/mL was added to primary cultures of fetal rat hippocampal neurons. The extract significantly elevated axonal length, number of secondary axonal collateral branches, length of primary dendrites and number of secondary dendritic branches by 58%, 8 folds, 68% and 2.6 folds, respectively, as compared to control [81].

Alzheimer’s disease is a common neurologic disorder, responsible for brain shrinkage and cell death. It is the most prevalent type of dementia and one of the top four causes of mortality in developed countries [83]. So far, the primary symptomatic therapy for this condition has been based on the “cholinergic hypothesis,” where the medications increase the level of acetylcholine in the brain by inhibiting the activity of the cholinesterase (acetylcholinesterase and butyrylcholinesterase) enzyme [84]. The anti-butyrylcholinesterase activity of a solvent extract of *D. dichotoma* has been reported. A methanol extract of *D. dichotoma* at a concentration of 1.3–6.5 mg/mL showed significant (*p* < 0.05) inhibition of cholinesterase enzyme (54.42%), compared to standard donepezil (cholinesterase inhibitor) (57.57%) at a concentration of 0.40–4.15 mg/mL [82]. 

The acetylcholinesterase and butyrylcholinesterase inhibitory activities of *C. racemosa* and *S. polycystum* at various concentrations (0.0125–0.2 mg/mL) have been determined. Solvent extracts of *C. racemosa* and *S. polycystum* indicated anti-acetylcholinesterase activities with IC_50_ values ranging from 0.086–0.115 mg/mL, while *C. racemosa* extracts indicated anti-butyrylcholinesterase activity with an IC_50_ value of 0.156 mg/mL [21].

A summary regarding the protective nature of the above-mentioned North Bornean seaweeds is shown in Table 2.

## 3. Nutraceutical Profiling of North Bornean Seaweeds

Seaweed nutraceutical bioactive compounds have great potential in biomedical and pharmaceutical applications [85,86,87,88,89,90,91,92]. Phytochemical analysis of Bornean seaweeds indicated the presence of various nutraceutical bioactive compounds with different properties, including phenolics, flavonoids, alkaloids, alcohols, steroids, fatty acids, terpenoids, etc. The nutraceutical bioactive compounds were identified in seaweed extracts by liquid chromatography–mass spectrometry (LCMS), gas chromatography–mass spectrometry (GCMS), and nuclear magnetic resonance (NMR) systems. The details are indicated in Table 3 and Figure 1a–g.

## 4. Methodology

The information was retrieved from multiple internet databases such as ScienceDirect, PubMed, Wiley, ACS publications, etc., and registers including theses and proceedings. Records were searched with keywords related to seaweed, North Borneo, distribution, taxonomy, bioactivity, secondary metabolites, and diseases. Around 250 records approximately from the year 2000 to 2021 were retrieved and screened. Among these, about 100 records were excluded due to being out of the scope of the review. Eventually, a total of 149 records were adopted for the current review paper, and data from organizations such as the World Health Organization were included as well.

## 5. Conclusions

In the current review, the protective effects of North Bornean seaweeds in terms of anti-inflammatory, antioxidant, antimicrobial, anticancer, anti-obesity and cardiovascular protection, neuroprotection, and renal protection as well as hepatic protection were described and followed by nutraceutical profiling. The protective roles of the seaweeds might be due to the presence of a wide variety of nutraceuticals including phthalic anhydride, 3,4-ethylenedioxythiophene, 2-pentylthiophene, furoic acid (*K. alvarezii*), eicosapentaenoic acid, palmitoleic acid, fucoxanthin, β-carotene (*E. denticulatum*), eucalyptol, oleic acid, dodecanal, pentadecane, (*H. durvillaei*), canthaxanthin, pentadecanoic acid, eicosane (*C. lentillifera*), pseudoephedrine, palmitic acid, monocaprin (*C. racemosa*), dictyohydroperoxide, squalene, fucosterol, saringosterol (*D. dichotoma*) and lutein, neophytadiene, cholest-4-en-3-one, and *cis*-vaccenic acid (*S. polycystum*). 

Despite their excellent pharmacological characteristics, only *K. alvarezii* and *E. denticulatum* are widely cultivated, developed as a functional food source, and used for carrageenans production. Locally, *C. lentillifera* and *C. racemosa* are also consumed as a nutrition source. For future perspectives, studies on functional food development and cultivation techniques are highly recommended. Furthermore, extensive studies on the seaweed isolates are needed to understand their bioactivity and mechanisms of action, while highlighting their commercialization potential.

## Figures and Tables

**Figure 1 marinedrugs-20-00101-f001:**
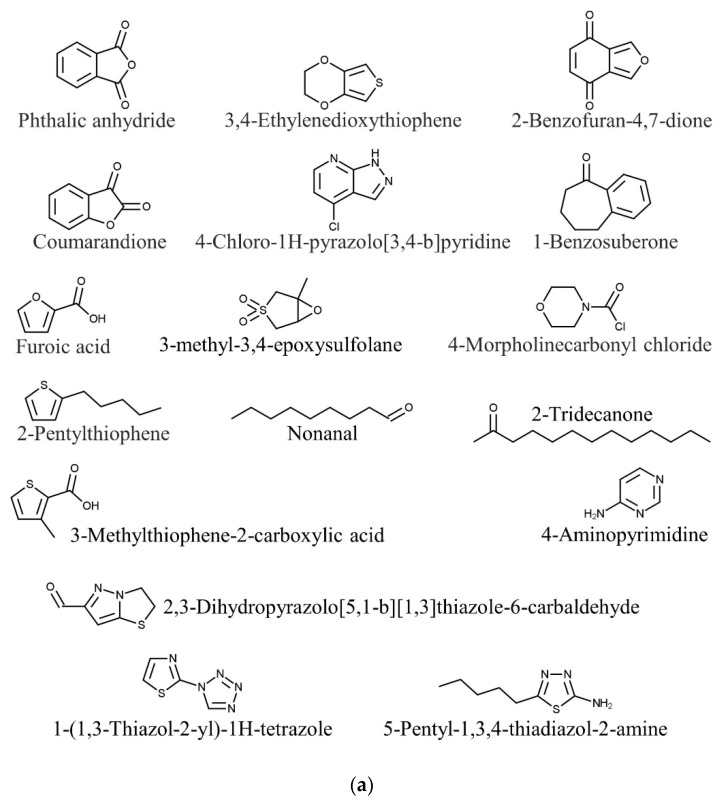
Phytochemical compounds of (**a**) *Kappaphycus alvarezii*, (**b**) *Eucheuma denticulatum*, (**c**) *Halymenia durvillaei*, (**d**) *Caulerpa lentillifera*, (**e**) *Caulerpa racemosa*, (**f**) *Dictyota dichotoma*, and (**g**) *Sargassum polycystum*.

**Table 1 marinedrugs-20-00101-t001:** Descriptions of North Bornean seaweed species.

Species	Phylum	Family	Other Names	References
*Kappaphycus alvarezii*	Rhodophyta	Solieriaceae	*Eucheuma cottonii* or sea bird nest	[9,10,11]
*Eucheuma* *denticulatum*	Rhodophyta	Solieriaceae	*Eucheuma spinosum*	[9,10,12]
*Halymenia durvillaei*	Rhodophyta	Halymeniaceae		[9,10]
*Dictyota dichotoma*	Ochrophyta	Dictyotaceae		[9,10]
*Sargassum polycystum*	Ochrophyta	Sargassaceae		[9,10]
*Caulerpa lentillifera*	Chlorophyta	Caulerpaceae	Sea grape	[9,10]
*Caulerpa racemosa*	Chlorophyta	Caulerpaceae	Sea grape	[9,10]

**Table 2 marinedrugs-20-00101-t002:** Summary of the protective effects of North Bornean seaweeds.

Seaweed Species	Protective Effects						
	Anti Inflammatory	Anti Oxidant	Anti Microbial	Anti Cancer	Anti-Obesity and Cardiovascular Protection	Hepato Protection	Neuro- Protection
*Kappaphycus alvarezii*	+	+	+	+	+	+	+
*Eucheuma denticulatum*		+	+	+	+		
*Caulerpa lentillifera*	+	+	+	+	+	+	
*Caulerpa racemosa*	+	+	+	+		+	+
*Halymenia durvillaei*		+	+	+			
*Dictyota dichotoma*		+	+	+			+
*Sargassum polycystum*	+	+	+	+	+	+	+

**Table 3 marinedrugs-20-00101-t003:** List of some important nutraceutical bioactive compounds identified in North Bornean seaweeds.

Species	Bioactive Compounds	Activity	References
*Kappaphycus alvarezii*	Phthalic anhydride	Antimicrobial	[85]
	1-Benzosuberone	Anticancer	
	3,4-Ethylenedioxythiophene	Antitumor	
	Furoic acid	Antimicrobial	
	Coumarandione	Antitubercular	
	2-Benzofuran-4,7-dione	Antimicrobial	
	2-Pentylthiophene	Antifungal	
	2,3-Dihydropyrazolo[5,1-b][1,3]thiazole-6-carbaldehyde	Anti-inflammatory, antibacterial, anticancer	
	1-(1,3-Thiazol-2-yl)-1H-tetrazole	Antimicrobial	
	4-Chloro-1H-pyrazolo[3,4-b]pyridine	Anticancer, antioxidant	
	4-Morpholinecarbonyl chloride	Antibacterial, antidepressant	
	3-Methylthiophene-2-carboxylic acid	Antibacterial	
	5-Pentyl-1,3,4-thiadiazol-2-amine	Antibacterial	
	3-Methyl-3,4-epoxysulfolane	Antifungal	
	Nonanal	Antifungal	[93,94]
	2-Tridecanone	Insecticide	[94,95]
	4-Aminopyrimidine	Neuroprotection	[96,97]
*Eucheuma denticulatum*	Docosahexaenoic acid	Anticancer, cardiovascular protection, anti-inflammatory	[86,87,88]
	Eicosapentaenoic acid	Anticancer, cardiovascular protection	[86,88,98]
	Palmitoleic acid	Anti-inflammatory	[86,99]
	Antheraxanthin	Antioxidant	[100,101]
	Astaxanthin	Anti-inflammatory, antioxidant	[100,102]
	*β*-Carotene	Antioxidant	[100,103]
	*β*-Cryptoxanthin	Anticancer, antioxidant	[100,104,105]
	Dinoxanthin	Antioxidant	[100,106]
	Diatoxanthin	Chemopreventive	[100,107]
	Diadinoxanthin	Antioxidant	[100,106]
	Fucoxanthin	Antitumor	[100,108]
	Lutein	Antioxidant	[100,106]
	Rubixanthin	Antioxidant	[100,109]
	Zeaxanthin	Antioxidant	[100,106]
*Halymenia durvillaei*	Eucalyptol	Antimicrobial, anti-inflammatory	[91,110,111]
	Caryophyllene	Antioxidant, anticancer, antimicrobial	[91,92]
	Palmitic acid	Anti-inflammatory, antioxidant, antiandrogenic 5- alpha-reductase inhibitor nematicide, pesticide, hypocholesterolemic	[17,91,112]
	Oleic acid	Quorum quenching and anti-biofilm potential	[91,113]
	Dodecanal	Antimicrobial	[91,114]
	Heptadecanoic acid	Anticancer	[91,115]
	Pentadecanoic acid	Anticancer	[91,116]
	Myristic acid	Larvicidal and repellent	[91,117]
*Caulerpa lentillifera*	Docosahexaenoic acid	Anticancer, cardiovascular protection, anti-inflammatory	[86,87,88]
	Eicosapentaenoic acid	Anticancer	[86,98]
	Palmitoleic acid	Anti-inflammatory	[86,99]
	Astaxanthin	Anti-inflammatory, antioxidant	[100,102]
	*β*-Carotene,	Antioxidant	[100,103]
	*β*-Cryptoxanthin	Anticancer, antioxidant	[100,104,105]
	Canthaxanthin	Antioxidant, anti-inflammatory, neuroprotection	[100,118]
	Zeaxanthin	Antioxidant	[100,106]
	Oleic acid	Antioxidant, cardiovascular protection, hepatic protection	[119,120]
	Pentadecanoic acid	Anticancer	[116,119]
	Myristic acid	Antidiabetic, anti-inflammatory	[119,121]
	Behenic acid	Anti-obesity	[119,122]
	Palmitic acid	Antitumor	[119,123]
	Limonene	Antiparasitic	[94,124]
	Heneicosane	Anti-inflammatory, analgesic, antipyretic	[94,125]
	Eicosane	Anti-inflammatory, analgesic, antipyretic	[94,125]
	Pentadecane	Anti-inflammatory, analgesic, antipyretic	[94,125]
	Azulene	Anti-inflammatory	[94,126]
*Caulerpa racemosa*	Pseudoephedrine	Anti-inflammatory	[90,127]
	Tetratetracontane	Antibacterial	[89,90]
	Deoxyspergualin	Nuclear factor-kappa B inhibitor	[90,128]
	2,4-Di-tert-butylphenol	Antibacterial, anti-inflammatory, anticancer	[129,130]
	Heptacosane	Antioxidant	[131]
	Palmitic acid	Anti-inflammatory, antioxidant	[123,132,133]
	Squalene	Anti-inflammatory, antioxidant, antitumor	[134,135]
	Methoxyacetic acid	Anticancer	[136,137]
		Effective against polycystic ovary syndrome (a hormonal dysfunction among women of reproductive age)	[136,138]
	Monocaprin	Antimicrobial	[136,139]
	D-Mannitol	Antihyperglycemic	[136,140]
*Dictyota dichotoma*	Dictyohydroperoxide	Anticancer	[141]
	Pentadecane	Anti-inflammatory. antiparasitic	[125,141,142]
	Squalene	Anti-inflammatory, antioxidant, antitumor	[134,135,141]
	Fucosterol	Anti-inflammatory, anticancer, hepatoprotective, antiphotoaging, anti-obesity, anti-Alzheimer’s disease, antioxidant	[141,143]
	Saringosterol	Anti-obesity	[141,144]
*Sargassum polycystum*	Docosahexaenoic acid	Anticancer, cardiovascular protection, anti-inflammatory	[86,87,88]
	Eicosapentaenoic acid	Anticancer, cardiovascular protection	[86,88,98]
	Palmitoleic acid	Anti-inflammatory	[86,99]
	Antheraxanthin	Antioxidant	[100,101]
	Astaxanthin	Anti-inflammatory, antioxidant	[100,102]
	*β*-Cryptoxanthin	Anticancer, antioxidant	[100,104,105]
	Canthaxanthin	Antioxidant, anti-inflammatory, neuroprotective	[100,118]
	Dinoxanthin	Antioxidant	[100,106]
	Diatoxanthin	Chemopreventive	[100,107]
	Diadinoxanthin	Antioxidant	[100,106]
	Fucoxanthin	Antitumor	[100,108]
	Lutein	Antioxidant	[100,106]
	Zeaxanthin	Antioxidant	[100,106]
	Palmitic acid	Anti-inflammatory, antioxidant, anti-androgenic 5- alpha-reductase inhibitor nematicide, pesticide, hypocholesterolemic	[17,112,132]
	Octadecenoic acid	Anticancer, antimicrobial	[17,112,145]
	Neophytadiene	Anti-inflammatory	[17,146]
	Myristic acid	Larvicidal and repellent	[117,147]
	*cis*-Vaccenic acid	Antiviral	[147,148]
	Cholest-4-en-3-one	Analgesic, neuroprotective, anti-obesity	[147,149]
	Squalene	Anti-inflammatory, antioxidant, antitumor	[135,147]

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
