# Peer review of "Therapeutic Potential and Nutraceutical Profiling of North Bornean Seaweeds: A Review"

_marinedrugs, 2022, doi:10.3390/md20020101_

Round 1

Reviewer 1 Report

I think it is a good review on the potential of algae as a healing remedy in human health. In this case it is macroscopic algae characteristic of warm seas. The bibliography is correct and sufficient, although it should change some small aspects.

  1) All Latin names of the species must be followed by the author, for this you can consult "The Catalogue of Life". 2) Rename Phaeophyta to Ochrophyta. 3) In table 1, follow this order: Rhodophyta -Ochrophyta - Chlorophyta.

4) Figure 1: improve the resolution of images F and G.

Author Response

Dear Editor,

Thank you for your valuable comments and suggestions which are very useful to improve our manuscript.

We have given our responses to all the queries raised by reviewer 2.

 Reviewer 2 comments

I think it is a good review on the potential of algae as a healing remedy in human health. In this case it is macroscopic algae characteristic of warm seas. The bibliography is correct and sufficient, although it should change some small aspects.

  • All Latin names of the species must be followed by the author, for this you can consult "The Catalogue of Life".

Res: Thank you very much for the comments all the Latin names of the species in the text have been shown along with the author.

  • Rename Phaeophyta to Ochrophyta.

Res: Phaeophyta has been renamed to Ochrophyta.

  • In table 1, follow this order: Rhodophyta -Ochrophyta - Chlorophyta.

Res: In table 1, the order: Rhodophyta -Ochrophyta – Chlorophyta has been followed.

4) Figure 1: improve the resolution of images F and G.

Res: The resolution of Figure 1 (F and G) has been improved.

Reviewer 2 Report

In their study, Muhammad Dawood Shah and colleagues reviewed Therapeutical Potential and Nutraceuticals profiling of North Bornean Seaweeds.  The following issues need to be addressed. 

  1. The title is confusing. ‘Disease management’ can be omitted.
  2. Search and screening strategy for literature is not defined. A PRISMA flow chart is recommended. Authors are advised to consult with Page at al. paper.

Page, M.J.; McKenzie, J.E.; Bossuyt, P.M.; Boutron, I.; Hoffmann, T.C.; Mulrow, C.D.; Shamseer, L.; Tetzlaff, J.M.; Akl, E.A.; Brennan, S.E.; et al. The PRISMA 2020 statement: An updated guideline for reporting systematic reviews. Journal of Clinical Epidemiology 2021, 134, 178-189, doi:10.1016/j.jclinepi.2021.03.001

  1. Future prospects should be clearly defined in a separate section before the conclusion or can be merged with conclusion.
  2. Line 55, polycystum will be in italic.
  3. Figure 1 should contain a measuring scale to compare different seaweed sizes.
  4. In section 2.1, Antiinflammatory needs hyphen.

Author Response

R1- Responses to the editor and reviewer 3 comments

Manuscript ID: marinedrugs-1543879.

 â€¯
Therapeutical Potential and Nutraceutical profiling of North Bornean Seaweeds: A Review

Dear Editor,

Thank you for your valuable comments and suggestions which are very useful to improve our manuscript.

We have given our responses to all the queries raised by reviewer 3.

Reviewer 3 comments

 Comments and Suggestions for Authors

In their study, Muhammad Dawood Shah and colleagues reviewed Therapeutical Potential and Nutraceuticals profiling of North Bornean Seaweeds.  The following issues need to be addressed. 

  • The title is confusing. ‘Disease management’ can be omitted.

Res: Thank you very much for the comments. The title has been reviewed and the term “Disease management” has been deleted. The new title in the text has been provided as “Therapeutical Potential and Nutraceutical profiling of North Bornean Seaweeds: A Review “

  • Search and screening strategy for literature is not defined.

Res: Thank you very much for the comments. The search and screening strategy for literature has been provided in the text as ‘’

The information has been retrieved from multiple internet databases such as Science Direct, PubMed, Wiley, ACS publications, etc and registers including thesis and proceedings. Records were searched with keywords related to distribution, taxonomy, bioactivity, secondary metabolites and diseases. Around 250 records approximately from the year, 2000 to 2021 were retrieved and screened. Among these, approximately 100 records were excluded as these are out of the review scopes. Eventually, a total of 149 records were adopted in the current review paper and data from organisations such as the world health organization were included as well.

  • Future prospects should be clearly defined in a separate section before the conclusion or can be merged with a conclusion.

Res:   A paragraph regarding the future prospects  has been merged with the conclusion and provided in the text as 

        “Despite their excellent pharmacological characteristics, only K. alvarezii and E. denticulatum were cultivated widely and being developed as a functional food source, aside for carrageenans production. Locally, C. lentillifera and C. racemosa are being consumed as well as a nutrition source. For future perspectives, studies on functional food development and cultivation techniques are highly recommended. Besides, extensive studies on the seaweed isolates are needed to understand their bioactivity action mechanisms, while highlighting their commercialization potential”.    

  • Line 55, polycystum will be in italic.

All scientific names have been rechecked thoroughly and corrected as recommended.

  • Figure 1 should contain a measuring scale to compare different seaweed sizes.

Res: Figure 1 has been displayed with a measuring scale bar in the text.

Round 2

Reviewer 2 Report

Authors addressed most of the comments, except comment#2 (Search and screening strategy for literature) which is partially responded.  It seems the authors could not follow the previous comment. However, authors added a search strategy at the end of Introduction which should be separated and be included as a 'Methodology'.  

Author Response

R2- Responses to the editor and reviewer 2 comments

Manuscript ID: marinedrugs-1543879.

 â€¯ Therapeutical Potential and Nutraceutical profiling of North Bornean Seaweeds: A Review

Dear Editor,

Thank you for your valuable comments and suggestions which are very useful to improve our manuscript.

We have given our response to the queries raised by reviewer 2.

Reviewer 2 comments

 Comments and Suggestions for Authors

Q: Authors addressed most of the comments, except comment#2 (Search and screening strategy for literature) which is partially responded.  It seems the authors could not follow the previous comment. However, authors added a search strategy at the end of Introduction which should be separated and be included as a 'Methodology'.  

Res: Thank you very much for the comment. The search strategy has been deleted from the introduction and included under Methodology.